# Sexual Citizenship Theory and Employment Discrimination among Transgender-Identified People

**Gina R. Rosich** 

Department of Social Work and Equitable Community Practice, University of Saint Joseph, 1678 Asylum Ave Rm 219, West Hartford, CT 06117, USA; grosich@usj.edu; Tel.: +1-860-2315-406

**Abstract:** Employment discrimination remains a consistent and widespread concern among transgender and gender non-conforming (GNC) people. A secondary data analysis was conducted using the Transgender Law Center California Economic Health Survey (n = 646). The aim of this study was to examine workplace discrimination among transgender and gender non-conforming adults. Sexual citizenship theory informed both the grouping of variables and analysis of findings. Bivariate, multivariate, and multivariable statistical tests were conducted to examine problems related to hiring and to various forms of workplace harassment. Analyses revealed that about 67% of respondents reported some kind of workplace mistreatment. Trans men (FtM) were 132.6% more likely to report discrimination in the workplace (chiefly misgendering and privacy breach), while trans women (MtF) were more likely to experience a wider variety of types of discrimination. Respondents out to their co-workers were 292.4% more likely to experiences discrimination. Those with higher income were less likely to need assistance with changing IDs and more likely to pass/blend. Those who were less likely to pass/blend faced higher unemployment. These findings underscore the many ways in which transphobia, cis gender entitlement and transmisogyny shape the lives of trans people and prohibit full citizenship participation in society vis-à-vis the workforce.

**Keywords:** employment discrimination; sexual citizenship theory; transgender; transmisogyny

## 1. Introduction

Transgender (trans) and gender non-conforming (GNC) people report being plagued by discrimination in the workplace, a reflection of mistreatment in society at large. Much of the literature on employment discrimination comes from community-based convenience samples and cross-sectional studies. These studies are largely atheoretical and descriptive in nature. Sexual citizenship theory, as an outgrowth of Marshall's theory of social citizenship, has much to offer when applied to the examination of the social, legal and economic injustices trans people face. It provides a useful framework for both analysis and advocacy. In this study, sexual citizenship theory is operationalized by bridging the core theoretical construct of civil, political and social society (the State) with the embodiment of trans identities under the pressures of cis gender entitlement, transmisogyny, hetero- and homonormativity. Many studies have provided atheoretical descriptive statistics of discrimination related to sociodemographic factors such as race, age, and education. This study fills a gap in the literature by applying sexual citizenship theory to specifically examine the effects of trans-specific factors such as gender presentation, passing/blending, being out as a trans person in the workplace, and personal identification documents (IDs). Adding to the multi-layered ways in which poverty, health, employment, personal safety, and access to other human rights impact a person's overall well-being, a constellation of sociodemographic factors in intersection with transgender-specific factors reveal insights into the ability of transgender adults to live economically independent and professionally rewarding lives [1–3].

## 1.1. Purpose of the Study

The purpose of this study was to examine workplace discrimination among transgender and gender non-conforming adults through the lens of sexual citizenship theory. In this study, various socioeconomic factors in intersection with the transgender-specific factors of gender identity, gender conformity in the workplace, passing/blending, personal identification documents (IDs), and being out in the workplace were used to examine and understand the lived experiences of discrimination by trans individuals. By using sexual citizenship theory, the hope is to illuminate the factors (both socioeconomic and trans specific) that identify who is most vulnerable to discrimination, and who is differentially vulnerable to particular types of discrimination.

## 1.2. Scope of the Problem

Employment discrimination is a pressing issue for people in the transgender community. The Williams Institute estimates there to be at least 700,000 adults in the United States who identify as transgender (0.3% of the adult population) [4]. An overwhelming 90% of respondents in the 2011 national survey of transgender discrimination reported experiencing harassment or mistreatment in the workplace [5]. In a 2010 study by Make the Road New York [6], 52% of the trans people they surveyed reported experiencing discrimination on the job while 49% reported receiving no employment offers while living as a trans person. NGLTF [7] reported 97% of respondents being mistreated or harassed at work. In total, 77% of employed respondents in the United States Transgender Survey (USTS) survey reported taking a variety of actions to avoid being mistreated in the workplace due to their gender identity or expression [8]. Transgender (trans) people and gender non-conforming (GNC) people experience disproportionately high rates of poverty, discrimination, disenfranchisement, and personal violence [3,5,8,9]. Descriptive studies have documented high rates of unemployment and underemployment as trans people encounter difficulties in hiring, on-the-job harassment, unfavorable work conditions, demotions, and termination of employment for reasons entirely unrelated to job performance—reasons such as personal bias, employer discomfort, and outright hostility towards trans people [10–12]. In the USTS, people of color from marginalized communities reported higher unemployment rates than their white counterparts [8].

This persistence of inequality and injustice has grave consequences for transgender and GNC individuals. Employment discrimination affects many areas of psycho/social functioning. Depression, anxiety, somatization, substance abuse and suicide are frequently reported outcomes when trans people are faced with chronic poor treatment, disenfranchisement, isolation, violence and discrimination [13–15]. Trans people are disproportionately negatively affected by discrimination in healthcare settings both in terms of access to and treatment by healthcare professionals. They face high rates of violence, and obstacles related to managing their physical and mental health, physical safety and legal recognition [2,16]. These areas are interrelated and create overlapping barriers to navigating everyday life. Furthermore, unemployment goes hand in hand with poverty, which puts trans people at risk for homelessness and resorting to sex work and other aspects of the underground economy as last options for survival. Trans people are disproportionately arrested and incarcerated (trans women in particular are often suspected of prostitution just for being trans), which further complicates their lives by exacerbating the problems of poverty and lack personal safety [1–3,17].

## 1.3. The Legal Landscape for Transgender Workers

In the United States, as of 2019, there are no Federal Legislative Protections for LGBTQ workers. Across the country, 22 States and over 150 individual cities and counties have explicitly trans-inclusive nondiscrimination employment laws on the books that apply to private employers [18]. In 2012, the Equal Employment Opportunity Commission held that employment discrimination based on gender identity, change of sex, gender stereotyping, and gender expression are cognizable forms of sex discrimination under Title VII of the Civil Rights Act of 1964. Trans people currently face threats

from the federal government. The Trump Administration has taken a series of steps to dismantle the rights of LGBTQ people. Some of these steps include instituting a military ban on transgender recruits and proposing rules that will allow religious organizations receiving federal government contracts the right to make employment decisions based on religious beliefs [19,20]. Crucially for trans workers, the Justice Department refuses to uphold the law protecting trans people in the workplace as established by the EEOC in the case of Macy v Holder and references to sexual orientation and gender identity protections have been removed from Department of the Interior and other Federal departmental websites [21].

*1.4. Sexual Citizenship Theory*

The intricacies of trans realities in the workplace can be understood and addressed through sexual citizenship theory. The approach taken in this study was to root it in Marshall's theory of social citizenship, and incorporate the concepts of heteronormativity, homonormativity, and transmisogyny into both the development of research questions and analysis of findings. Discussion of these concepts will first start with social citizenship theory.

Marshall's [22] theoretical framework of social citizenship is a tripartite approach comprised of (a) civil society, (b) political society, and (c) the State (also known as social society). Civil society encompasses those societal forces in both everyday communal life and economic life where one's membership is initially defined. It includes the right to liberty of person, freedom of speech, thought and religious faith, the right to own property, to personal safety, to make and uphold valid contracts, and the right to seek justice through due process of law in the courts. It also means the right to economic welfare and access to systems such as education and social services. Marshall explicitly includes among these the right to work in an occupation of one's choice based on one's ability to meet the skills and educational demands required for the position. Political society encompasses democratic representation, including the right to exercise political power as an elected official and the right to vote for elected officials who will represent one's interest in the passage of laws that reflect the political interests of the voter. The State (also known as social society) encompasses the delivery of social responsibilities through various systems such as public education, economic welfare (e.g., wage regulations and interest rates), and social services (e.g., public assistance). In this study, social society will also include the legal recognition of personhood. Cyclically, civil rights give way to political power and political power curtails civil prejudice. The State employs mechanisms that make it possible for members to exercise their civil and political rights. Full citizenship represents the promise of equal value in society, and the lack of equality can be explained and predicted when the promise of citizenship is denied in one or all areas. Disenfranchisement in any one of the three spheres of the framework can tip the balance against a group. Breiner [23] pinpoints the effects of capitalism as the source of exclusionary conflict. He describes Marshall's theory as a "relentless struggle" between the principles of political and social inclusion and capitalist forces pushing back through civil exclusion. The lack of economic opportunity, as denied through civil society, becomes the primary arena for struggle, pushing back the ability of oppressed people from gaining or exercising political and social rights.

In this framework, trans people struggle with the ability to participate in the workplace due to exclusions in all three realms. First is the struggle to obtain and maintain employment in a safe and supportive atmosphere. There is an ongoing battle for laws that represent the interests of trans people or elected officials to prioritize their concerns. Legal recognition by the State of one's gender and personhood are met with resistance. Bureaucratic and gatekeeping steps are created to prevent people from changing the gender markers on their identity documents to accurately reflect their gender identity and expression [24]. Together, this means trans people must navigate the private choices of transition when the right to bodily autonomy is questioned in the most fundamental of legal terms. The public embodiment of gendered bodies in institutional, social and legal spaces becomes a navigational minefield for surviving the workplace.

Societal membership is predicated upon social acceptance in the community such that the shared identity of the community allows for considering a particular subgroup as citizens worthy of social inclusion, dignity and respect [25,26]. When someone is viewed as an "other," their exclusion sends the message "you don't belong." When a person's existence occurs outside the comfort zone or threshold of societal norms, this can lead to either full exclusion or what might be referred to as marginal citizenship. In marginal citizenship, full enjoyment of inclusion cannot be obtained, but partial participation is still possible. Full exclusion entails the loss of the right to even have rights, or of being viewed as a valued member of society [26,27].

Richardson [25] examined citizenship and sexual minorities, arguing that citizenship status is closely linked with male privilege and institutionalized heterosexuality (heteronormativity). Lesbians and gays, in her model, have a limited ability to exercise political power. Building on her model, trans people are similarly judged and excluded due to a combination of homonormativity and cis gender entitlement. If heteronormativity is the universalization of heterosexuality in society, homonormativity is the framing of same-sex relationships in a normalizing fashion. By using assimilationist terms and images that prioritize white, middle class "straight acting" lesbians and gays as an ideal, homonormativity is the act of "normalizing" same-sex relationships on heteronormative terms, embracing gender binaries by eschewing the stereotypes that all gay men are effeminate and all lesbians are masculine. This leads to the exclusion of gender non-conforming people for the sake of the social and political inclusion of lesbians and gays. It also sets the stage for a standard of gender recognition based on an ideal of gender presentation known as passing [28]. Passing privilege is the notion that it is preferable to not appear different or somehow stand out as gender transgressive. It is often determined by cis gender people who view passing as an achievement, and who set the bar for what is considered acceptable gender privilege. It is conditional in that once someone comes out or is outed as trans, they are no longer seen as their authentic gendered self and are sometimes accused of deceiving others [29,30].

### 1.5. Research Questions

To gain a dynamic understanding of the workplace problems faced by trans people that goes beyond the important work of documenting the prevalence of the problem, this study asked the following questions: How do gender identity, presentation (including IDs), passing/blending and conformity to the gender binary impact the likelihood of discrimination in the workplace? Who is most likely to experience the problems of workplace discrimination and unemployment/underemployment? What are the potential consequences (both positive and negative) of being out of work? Who is most likely to experience particular types of workplace discrimination?

## 2. Materials and Methods

### 2.1. Participants

This study was a secondary data analysis using the quantitative portion of the Transgender Law Center's (TLC) California Transgender Economic Health Survey (CTEHS), a cross-sectional statewide study of transgender individuals. Human subjects' approval for the initial study was obtained through the California State University. The author of this study obtained IRB approval for the secondary data analysis through Fordham University. The CTEHS was a retrospective, cross-sectional descriptive and associational study with quantitative and qualitative components, and this subsequent analysis is both descriptive and associational. The target population and inclusion criteria for this study were all individuals in the state of California, age 18 and over who identify as transgender or gender non-conforming. A total of n = 646 respondents were included in the final analysis. Response rates were not available. Participants were recruited using convenience and snowball sampling methods. The survey was distributed in English and Spanish both in print and electronically (through SurveyMonkey).

*2.2. Measures*

2.2.1. Sociodemographic Variables

The sociodemographic questions of age, income, and race were examined in this study. Income was measured by asking respondents their personal income before taxes in 11 grouped categories. This was then recoded into two categories (above and below $20k) as a proxy for those living at or below the poverty line and those living above the poverty line. Race was determined by asking respondents to self-identify in one of eight categories. This was then recoded into five categories. To reduce cell size, it was also reduced into two categories (white/person of color) for some tests.

2.2.2. Transgender-Specific Sociodemographic Variables

To reach for the complexity of the transgender experience, the researchers asked questions specifically designed to examine their gendered lives. For example, rather than asking if someone was male or female participants were asked what was their sex assigned at birth. They were also asked if they considered themselves transgender in any way or a gender different than that assigned at birth. These variables were then transformed so that anyone who said yes and was assigned male at birth became "MtF" and anyone who said yes and was assigned female at birth became "FtM". Respondents were also asked if they currently have a gender identity or presentation that differs from the sex assigned at birth, and were given a choice of 13 different options for terms used to describe their gender identity including (but not limited to) cross-dresser, feminine male, genderqueer, masculine female or butch, and two-spirit, in addition to FTM/transgender man and MTF/transgender woman. Anyone who answered that they used the terms "FtM/transgender man" or "MtF/transgender woman" was put in the corresponding category FtM or MtF. Gender conformity in the workplace was determined by asking respondents, "How do you present your gender in the workplace?" in six categories. This was conceptually recoded into two categories: gender conforming and gender non-conforming for cell-size reasons and to perform certain statistical analyses.

2.2.3. Passing/Blending, and Being out at Work

Passing/blending was determined by asking respondents, "When people meet you for the first time, how often do they guess you are transgender or gender non-conforming?" in five ordinal categories. This was then recoded into two categories to reduce cell size for some tests such that "never" and "infrequently" = no and "sometimes," "more often than not" and "always = yes.

Being out at work (out) was determined by asking separately if they were out to co-workers and out to their boss (out to boss). Respondents were asked, "About how many of your co-workers know that you are transgender?" in four ordinal categories. This was then recoded for some tests into two categories where "less than half," "almost half," and "all" = yes and "none of them" = no. Respondents were then asked to answer yes or no to the question "Does your boss or supervisor know you are transgender?"

2.2.4. Discrimination, Documents, and Unemployed (for Being Trans)

Types of employment discrimination were examined by asking respondents to check off any and all options that applied to them from a list of workplace experiences. The options listed were particular to transgender and gender non-conforming individuals, and experienced by the respondents specifically because of being transgender or gender non-conforming. The discrimination outcome categories were job loss (including fired, denied a promotion, laid off, or reorganized out of a job), harassment/violence (including verbal or sexual harassment, harassed by co-workers or supervisors, or victim of physical violence), marginalized (faced unfair scrutiny/discipline or restriction/elimination of access to clients and/or customers), bathrooms (denied access to appropriate restrooms or restrooms that matched my gender or denied access to all bathrooms), misgendered (supervisors or co-workers repeatedly used old name/pronoun even after being corrected), privacy breach (co-workers or supervisors shared

information that they should not have), and none. This multi-option question was also recoded into a dichotomous variable (yes/no) to test the global experience of discrimination in some statistical tests.

Participants were also asked if they were interested in assistance with their career in a variety of categories. The option "Changing documents to match my gender identity" was used as a proxy in this study for someone whose identifying documents did not match their gender identity at the time of the study and will be referred to as IDs in the remainder of this paper. Unemployed (for being trans) was determined by asking respondents "Have you ever been unemployed as a result of being transgender or gender non-conforming."

*2.3. Analytic Approach*

Transforming the variable of types of discrimination in the workplace into a binary variable (yes/no) allowed for the testing of global experiences of discrimination. Multinomial logistic regression, however, allowed for the unpacking of discrimination in the workplace to take a more nuanced look at the data to see who was more likely to experience particular types of discrimination. Multinomial logistic regression builds upon the method of logistic regression by allowing the researcher to create a model including a series of independent binary logistic regressions so as to better utilize categorical variables where continuous variables are either not available or do not make sense [31]. In this case, there is no scale of discrimination. Trans people report a variety of types of discrimination in the workplace that cannot be measured as worse than another. But by taking a detailed look at the data, researchers and advocates can better understand the dynamics at play and how to advocate and educate for more equitable treatment in the workplace.

All variables were examined to assess the amount of missing data. The distribution of variables was examined to determine that they met the underlying assumptions of statistical tests. Logistic models were checked to determine that all assumptions of logistic regression were met.

2.3.1. Bivariate Analyses

Based on the level of measurement of the variables, chi-square, correlation, 1-way ANOVA and independent sample *t*-tests were conducted to test the association between each of the variables, with covariates and the moderating variable. Chi-square was conducted to test the relationship between gender and types of discrimination for possible transmisogyny. Chi-square was also conducted to examine the variable of income in relation to the variables passing/blending and IDs, which are common sources of struggle during transition. Finally, chi-square was conducted to test the relationship between being unemployed and the independent variables gender, gender conformity, and passing/blending for possible transmisogyny and cis gender entitlement.

2.3.2. Multivariable and Multivariate Analyses

Logistic regressions were performed to test the relationship between the global experience of discrimination and the independent variables of gender, gender conformity, out to boss, out to co-workers, income, race and IDs to see if misogyny/transmisogyny or being out makes one more likely to experience discrimination. Two logistic regressions with interaction effect were performed. The first was if interaction effect of being out and documents (IDs) increased the likelihood of experiencing discrimination. The second was if interaction effect of being to boss, IDs, and passing increased the likelihood of experiencing discrimination. As a concept, this created the latent variable for testing the compounded effects of these variables within the framework of citizenship theory. See Table 1.

A hierarchical logistic regression was performed to test whether income modified the relationship between experiencing discrimination in the workplace and gender, and between experiencing discrimination in the workplace and the interactions of out to boss and IDs, and out to boss, IDs, and passing/blending. Finally, multinomial logistic regressions were performed to examine who was more likely to experience specific types of discrimination with the independent variables gender, gender conformity, passing/blending, IDs, out to co-workers, out to boss, and age category.

**Table 1.** Variables within the Theoretical Framework (both dependent and independent).

| Civil Realm | Political Realm | Social Realm |
|---|---|---|
| • Gender (MtF/FtM)» <br> • Passing/blending» <br> • Being Out to Co-workers <br> • Being Out to the Boss <br> • Gender Conformity in the Workplace» <br> • Unemployed (for being trans)» <br> • Global experience of discrimination <br> • Specific Types of Workplace Discrimination Experienced to include: <br>   ○ job loss <br>   ○ harassment/violence <br>   ○ marginalization <br>   ○ access to bathrooms <br>   ○ misgendering <br>   ○ privacy breach | • Income <br> • Race | • Reported need of assistance with obtaining documents (IDs) » |

» = Variables were also used to examine the concepts of transmisogyny or cisgender entitlement.

## 3. Results

### 3.1. Descriptives

Table 2 shows the socioeconomic demographics. The sample consisted of 646 adults, of which 58.3% identified as MtF (trans women) and 41.7% identified as FtM (trans men). Respondents ranged in age from 18 to 74. The mean age was 39, and the median age was 37 (SD 12.9). An independent *t*-test showed the mean age for trans women was 42.6, which was significantly older than trans men at 34.4, t (599) = 8.232, $p \leq 0.001$. Most respondents were white (63.2%), while 36.8% were persons of color with a breakdown of 5.8% black, 5.1% Asian/Pacific Islander (API), 13.3% Latinx and 12.6% other/multiracial. In total, 41% of respondents had income below $20,000 a year, and 59% had income above $20,000 with a breakdown of $20,000 to $49,000 = 30.1%; $50,000 to $69,000 = 11.1%; $70,000 to $99,000 = 8.8%; and $100,000 or more = 9%. In terms of education, 13.7% reported having a high school degree or less. In total, 40% had some college, 17% had a college degree, and 29.3% reported graduate education.

**Table 2.** Socioeconomic Demographics.

| Socioeconomic Demographics | N = 646 | | | |
|---|---|---|---|---|
| Age | Range <br> 18–74 | Mean <br> 39 | Median <br> 37 | SD <br> 12.9 |
| Socioeconomic Demographics | Percentage | | | |
| Gender <br> MtF <br> FtM | <br> 58.3% <br> 41.7 | | | |
| Race <br> White <br> PoC <br> Black <br> API <br> Latino <br> Other/multiracial | <br> 63.2% <br> 36.8% <br> 5.8% <br> 5.1% <br> 13.3% <br> 12.6% | | | |

**Table 2.** *Cont.*

| Socioeconomic Demographics | N = 646 |
|:---:|:---:|
| Income | |
| Below $20,000 | 41% |
| Above $20,000 | 59% |
| $20,000–$49,000 | 30.1% |
| $50,000–$69,000 | 11.1% |
| $70,000 to $99,000 | 8.8% |
| $100,000 + | 9% |
| Education | |
| High School or Less | 13.7% |
| Some College or More | 86.3% |
| Some College | 40% |
| College Degree | 17% |
| Graduate Education | 29.3% |

### 3.1.1. Transgender-Specific Descriptives

Table 3 shows the transgender-related demographics. Slightly more than half (53.3%) of respondents self-reported as passing. Most respondents (78.5%) reported as gender-conforming at work. The majority of respondents (76.5%) reported they were out to their co-workers, and 74.7% were out to their boss. About 67% of respondents reported some form of discrimination in the workplace. Less than half of respondents (35.9%) reported being unemployed because they were trans. Only 36.4% of respondents indicated needing help with changing their identifying documents.

**Table 3.** Transgender-related Demographics.

| Transgender-Related Demographics | Yes | No |
|:---:|:---:|:---:|
| Passing/blending | 53.3% | 46.7% |
| Gender Conformity at work | 78.5% | 21.5% |
| Out to Co-workers | 76.5% | 23.5% |
| Out to their Boss | 74.7% | 25.3% |
| Unemployed for being transgender | 35.9% | 64.1% |
| Experienced Discrimination at Work | 67% | 33% |
| Need help changing documents/IDs | 36.4% | 63.6% |
| Types of Discrimination Reported | Percentage | |
| Privacy Breach | 30.8% | |
| Misgendered | 9.6% | |
| Harassment/Violence | 8.6% | |
| Marginalized | 8.2% | |
| Job Loss | 6.2% | |
| Bathroom Issues | 3.3% | |

### 3.1.2. Bivariate Findings for Discrimination

In relation to the dependent variable of discrimination (global), all independent variables except for gender conformity in the workplace and passing/blending were statistically significant. Among the total number of respondents who reported being discriminated against, 55% were MtF and 45% were FtM. $\chi^2$ (1, N = 554) = 4.17, $p < 0.05$ Within groups, 63.9% of MtF respondents and 72.1% of FtM respondents reported being discriminated against. There was a significant association between being out to one's co-workers and experiencing discrimination. An overwhelming 84.5% of those discriminated against were out to their co-workers while only 15.5% of those discriminated against were not out to their co-workers. $\chi^2$ (1, N = 352) = 15.94 $p < 0.001$ Similarly, 80.8% of respondents who reported discrimination were out to their bosses, while 19.2% were not out. $\chi^2$ (1, N = 364) =

12.01 $p \leq 0.001$. Among respondents who reported being discriminated against, 60% did not check off needing assistance with documents and are assumed to have IDs with the correct name and gender markers consistent with their gender identity. $\chi^2$ (1, N = 582) = 4.747 $p < 0.05$. Two variables showed no significance testing.

When examining the relationship between gender and the expanded dependent variable of discrimination (Figure 1), trans men were more likely to experience misgendering, while trans women were more likely to report a wider range of types of discrimination including job loss, harassment/violence, and bathroom access problems. $\chi^2$ (6, N = 554) = 29.726, $p < 0.001$.

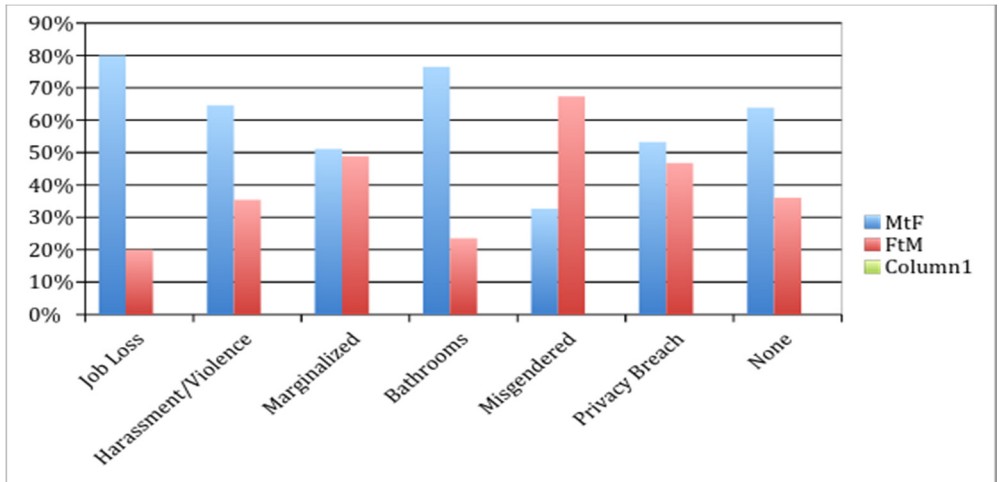

**Figure 1.** Types of Discrimination by Gender. $\chi^2$ (6, N = 554) = 29.726, $p \leq 0.001$.

### 3.1.3. Bivariate Findings for Income and Unemployed for Being Transgender

There was a weak negative relationship between passing and income $r_s$ (597) = −0.102, $p < 0.05$. Those with less income were less likely to pass. Conversely, those who do not pass were likely to have less income. Additionally, there was a statistically significant relationship between income and needing assistance with documents $\chi^2$ (1, N = 602) = 14.726, $p \leq 0.001$. Needing assistance with documents was associated with lower income. In relation to the dependent variable unemployed for being trans, only passing/blending was statistically significant. $\chi^2$ (1, N = 529) = 9.911, $p \leq 0.002$. While trans women were more likely to report being unemployed for being trans then trans men (64.8% MtF versus 35.2% FtM), these differences were not statistically significant at the 0.05 level.

### 3.1.4. Multivariate and Multivariable Findings

Logistic regression was employed to predict the probability that a participant would experience discrimination in the workplace. The predictor variables were gender, gender conformity, being out to your boss, out to co-workers, and indicating needing help with documents (IDs). In Model 1, gender, out to co-workers and needing help with IDs were significantly related. The model was checked to determine that all assumptions of logistic regression were met. A test of the full model compared to the intercept-only model was statistically significant (5, N = 308) = 34.097, $p < 0.001$. The model was able to accurately classify with 66.9% accuracy. This was an improvement of 5.5% from the intercept only model. Table 4 Model 1 shows the logistic regression coefficient, Wald test and odds ratio for each of the predictors. The odds ratios indicate that trans men (FtM) were 132% more likely to experience discrimination in the workplace than trans women (MtF). Those who were out to co-workers were 292.4% more likely to experience discrimination in the workplace than those who were not. Those who indicated needing help with IDs were 98.9% more likely to experience discrimination in the workplace than those who did not.

**Table 4.** Logistic Regression: Likelihood of Experiencing Discrimination in the Workplace.

| | Model 1 | | | | Model 2 with Interaction Effect | | | | Model 3 Hierarchical | | | |
|---|---|---|---|---|---|---|---|---|---|---|---|---|
| | Discrimination Y/N | | | | Discrimination Y/N | | | | Discrimination Y/N | | | |
| IVs | β | Wald | *p* | OR | β | Wald | *p* | OR | β | Wald | *p* | OR |
| Gender | 0.844 | 10.254 | **0.001** | 2.326 | | | | | 0.650 | 7.647 | **0.006** | 1.915 |
| Gender Conforming | −0.513 | 2.597 | 0.107 | 0.599 | | | | | | | | |
| Out to Boss | −0.084 | 0.037 | 0.848 | 0.920 | | | | | | | | |
| Out to Co-workers | 1.367 | 8.965 | **0.003** | 3.924 | | | | | | | | |
| Help w/IDs | 0.688 | 6.773 | **0.009** | 1.989 | | | | | | | | |
| Passing | | | | | | | | | | | | |
| Out to Boss * IDs | | | | | 0.907 | 6.628 | **0.010** | 2.477 | 0.813 | 6.697 | **0.010** | 2.256 |
| Out to Boss * IDs * Passing | | | | | −0.105 | 0.053 | 0.819 | 0.900 | | | | |
| Passing * IDs | | | | | | | | | 0.150 | 0.164 | 0.686 | 1.161 |
| Income | | | | | | | | | −0.057 | 0.051 | 0.821 | 0.944 |
| −2LL | 376.833 | | | | 466.711 | | | | 435.635 | | | |
| | $\chi^2$ (5, N=308) =34.097, $p < 0.001$ | | | | $\chi^2$ (2, N = 362) =11.488 $p \leq 0.003$ | | | | $\chi^2$ (4, N = 344) = 17.428, $p \leq 0.002$ | | | |
| Nagelkerke $R^2$ | 0.142 | | | | 0.043 | | | | 0.067 | | | |
| Hosmer & Lemeshow | 0.862 | | | | 1.000 | | | | 0.745 | | | |
| Classification Accuracy | 66.9% | | | | 62.7% | | | | 64% | | | |

*p* values for significant relationships are shown in bold.

Table 4, Model 2, shows the logistic regression employed with interaction effect to predict the probability that a participant would experience discrimination in the workplace. The interaction effects examined were being out to your boss by needing help with IDs, and being out to your boss by needing help with IDs by passing. The model was checked to determine that all assumptions of logistic regression were met. Table 4 shows the logistic regression coefficient, Wald test and odds ratio for each of the predictors. A test of the full model compared to the intercept-only model was statistically significant $\chi^2$ (2, N = 362) = 11.488 $p \leq 0.003$. The model was able to accurately classify with 62.7% accuracy. The odds ratios indicate that respondents were 147.7% more likely to experience discrimination in the workplace if they were both out to their boss and indicated needing help with IDs. There was no significance to the interaction of being out to your boss by help with IDs by passing.

Table 4, Model 3, shows the final model of a hierarchical logistic regression employed to predict the probability that income would moderate the relationship between experiencing discrimination in the workplace and the variable gender, and the interaction variables being out to boss by IDs and IDs by passing. The models were checked to determine that all assumptions of logistic regression were met. A test of the model without the modifier of income was statistically significant. In the final model, the odds ratio indicated that trans men (FtM) were 91.5% more likely to experience discrimination than trans women. The interaction of out to boss and IDs indicated those who were out and indicated needing assistance with their IDs were 125.6% more likely to be discriminated against. Income did not moderate these relationships. The interaction of passing by IDs was not significant in either block of the model.

Table 5a–c show the final models of multinomial logistic regressions run to predict the probability that gender, gender conformity, passing/blending, being out to one's boss or co-workers, age, race or needing assistance with IDs would impact the likelihood of experiencing specific types of discrimination in the workplace. The regression coefficient, Wald test, and odds ratio for each of the predictors are shown. Due to the number of possible outcomes (8), four regressions were run with two independent variables each. This was done due to the small cell size resulting from the multiple outcome possibilities

in the dependent variable. Having too many predictor variables (IVs) in the model resulted in errors associated with overfitting the model (Frost, 2015). The models were checked to determine that all other assumptions of multinomial logistic regression were met. The most frequently reported type of discrimination was privacy breach (30.8%), and the least reported type of discrimination was bathrooms (3.3%).

**Table 5.** (**a**) Gender and IDs by Types of Discrimination. (**b**) Age and Out to co-workers by Types of Discrimination; (**c**) Race and Out to Boss by Types of Discrimination.

| (a) | | | | | | | | |
|---|---|---|---|---|---|---|---|---|
| | **Gender** | | | | **Need Assistance with Documents (IDs)** | | | |
| | β | Wald | *p* | OR | β | Wald | *p* | OR |
| Misgendered | 1.351 | 16.860 | ≤0.001 | 3.861 | 0.657 | 4.196 | <0.05 | 1.929 |
| Privacy | 0.488 | 4.957 | <0.05 | | 0.515 | 5.352 | <0.05 | 1.673 |
| Reference Category | "None" | | | | | | | |
| Model Fitting Information | <0.001 | | | | | | | |
| N | 554 | | | | | | | |
| Nagelkerke R$^2$ | 0.076 | | | | | | | |
| (Goodness-of-fit) Pearson | 0.858 | | | | | | | |
| (Goodness-of-fit) Deviance | | | | 0.743 | | | | |
| (b) | | | | | | | | |
| | **Age** | | | | **Out to Co-Workers** | | | |
| | β | Wald | *p* | OR | β | Wald | *p* | OR |
| Types of Discrimination | | | | | | | | |
| Job Loss | | | | | | | | |
| Harassment/Violence | −0.047 | 4.697 | <0.05 | 0.954 | | | | |
| Marginalized | | | | | 1.363 | 4.522 | <0.05 | 3.909 |
| Bathrooms | | | | | | | | |
| Misgendered | −0.056 | 10.485 | ≤0.001 | 0.945 | 1.505 | 7.028 | <0.01 | 4.505 |
| Privacy | −0.024 | 4.678 | <0.05 | 976 | 1.232 | 12.892 | ≤0.001 | 3.429 |
| Reference Category | | | | "None" | | | | |
| Model Fitting Information | | | | <0.001 | | | | |
| N | | | | 346 | | | | |
| Nagelkerke R$^2$ | | | | 0.116 | | | | |
| (Goodness-of-fit) Pearson | | | | 0.966 | | | | |
| (Goodness-of-fit) Deviance | | | | 1.000 | | | | |
| (c) | | | | | | | | |
| | **Race** | | | | **Out to Boss** | | | |
| | B | Wald | *p* | OR | β | Wald | *p* | OR |
| Types of Discrimination | | | | | | | | |
| Job Loss | | | | | | | | |
| Harassment / Violence | | | | | | | | |
| Marginalized | | | | | 1.136 | 3.928 | <0.05 | 3.113 |
| Bathrooms | | | | | | | | |
| Misgendered | | | | | 1.443 | 6.618 | ≤0.01 | 4.232 |
| Privacy | −0.551 | 3.974 | <0.05 | 0.576 | 1.078 | 1.279 | ≤0.001 | 2.938 |
| Reference Category | | | | "None" | | | | |
| Model Fitting Information | | | | < 0.01 | | | | |
| N | | | | 360 | | | | |
| Nagelkerke R$^2$ | | | | 0.081 | | | | |
| (Goodness-of-fit) Pearson | | | | 0.271 | | | | |
| (Goodness-of-fit) Deviance | | | | 0.169 | | | | |

Only significant relationships are shown.

Multinomial logistic regressions were employed to predict the probability that the participant would experience particular types of discrimination. In the first model (Table 5a), gender and indicating need of assistance with IDs were used as the basis for comparison and "none" was the reference category for possible DV outcomes. The Nagelkerke Pseudo R-square for the model indicates 7.6% of the variance was explained by the model. Trans men were 286% more likely to report being misgendered, and those indicating a need of assistance with IDs were 92.9% more likely to report being

misgendered. Trans men were 63% more likely to report a privacy breach. Those indicating a need of assistance with IDs were 67.3% more likely to report a privacy breach. No other outcome categories were significant.

In the second model (Table 5b), age and out to co-workers were used as the basis for comparison and "none" was the reference category for possible DV outcomes. The Nagelkerke Pseudo R-square for the model indicates 11.6% of the variance was explained by the model. As age increased, respondents were 4.6% less likely to report harassment. Those out to co-workers were 290.9% more likely to report being marginalized. As age increased, respondents were 5.5% less likely to be misgendered. Those out to co-workers were 350.5% more likely to be misgendered. As age increased, respondents were 2.4% less likely to report a privacy breach. Those out to co-workers were 242.9% more likely to report a privacy breach. No other outcome categories were significant.

In the third model (Table 5c), race and out to boss were used as the basis for comparison and "none" was the reference category for possible DV outcomes. The Nagelkerke Pseudo R-square for the model indicates 8.1% of the variance was explained by the model. Those out to their boss were 211.3% more likely to experience being marginalized, 323.2% more likely to be misgendered, and 193.8% more likely to report a privacy breach. People identifying as white were 42.4% less likely to report a privacy breach. No other outcome categories were significant.

There were no significant outcomes for the independent variables gender conformity or passing. This is consistent with, and flows from, the bivariate analyses for these independent variables and the likelihood of experiencing discrimination.

Tables 6 and 7 show the relationships between income and the variables passing and IDs, which were tested via logistic regressions. In the relationship between passing and income, the likelihood of passing decreased by 17.2% for each decrease in income unit. B $-0.188$, $p < 0.01$ OR 0.828. In the relationship between needing assistance with IDs and income, the likelihood of $\beta$ $-0.295$, $p < 0.001$ OR 0.744. As income increased, the need for assistance with IDs decreased by 25.6% for each decrease in income unit.

**Table 6.** Relationship between Income and Passing.

| | **Passing Y/N** | | |
|---|---|---|---|
| | Correlation | | |
| | Observed | $r_s$ | $p$ |
| Income | 597 | $-0.102$ | **0.05** |
| | Logistic Regression | | |
| | $\beta$ | Wald | $p$ | Exp(B) |
| Income | $-0.188$ | 8.255 | **0.004** | 0.828 |

**Table 7.** Relationship between Income and Needing Assistance with Documents (IDs).

| | **Documents Y/N** | | |
|---|---|---|---|
| | Chi Square | | |
| | N | <u>df</u> | $\chi^2$ | $p$ |
| Income | 602 | 1 | 14.726 | **0.001** |
| | Logistic Regression | | |
| | $\beta$ | Wald | $p$ | Exp(B) |
| Income | $-0.295$ | 16.697 | **0.001** | 0.744 |

3.1.5. Statistical Power

Statistical power was determined using GPower. The power ranged from 0.800 to 0.844, indicating a low probability of Type II error. Failure to find significant associations between specific types of discrimination and the variables gender and gender conformity should be made with some caution, however, as some of the outcomes had an N less than 15, and one of the outcomes (violence) had an N less than 10.

## 4. Discussion

A commonly held belief in the trans community is that "The key to the world treating you well as trans is passing" (Jayna Pavlin of Trans-ponder podcast, Personal Communication). Yet in this study, the variable passing/blending was only significantly attributed to being unemployed. This, then, may be less of a factor for those already employed, and more of one for those looking for work, (some of whom may be in the process of transitioning). The 2010 matched pair testing study [6] highlighted difficulties trans people face in seeking gainful employment when qualifications are not at issue.

Passing/blending and job seeking were both poverty-related in this study. Poverty and unemployment are well-documented problems faced by members of the trans community and with 41% of respondents in this study reporting annual incomes under $20k, this sample was no exception [5,32,33]. Job-seeking requires the availability of money to cover costs such as transportation and interview outfits. Add to that the passing-related costs of grooming (e.g., haircuts, hair removal, possibly an entirely new wardrobe of gender-confirming workplace attire), and transition-related medical healthcare that may go unfunded such as hormones and surgeries [2,34], make it easy to see how passing becomes a poverty issue. Since the likelihood of passing decreased as income decreased in this study, together these factors of passing, unemployment and poverty suggest there is a vicious circle of prolonged poverty that keeps some trans people from obtaining or maintaining employment and financial security. Rejection that is fueled by standards of appearance established through cis gender entitlement rather than work qualifications results in gatekeeping of a fundamental aspect of societal participation.

Whereas lack of income presented problems, increased income was inversely associated with a need for help with IDs. This suggests that some of the problems faced with changing personal identifying documents may be mitigated by increased income. This could include the ability to hire a lawyer to navigate the legal systems; having a salaried position with benefits that include the ability to take paid personal/vacation days in order to spend the time updating one's paperwork; or the access to health care insurance or funds to cover transition-related medical costs. Medical (including surgical) transition has been a common requirement as a precursor to changing IDs under gatekeeping policies reflecting a cis gender entitled medical model of transition [2,29,35].

In the workplace, problems were more likely to occur when others became aware that someone was trans, rather than based in standards of passing or conformity to cis gender standards of gender presentation within the gender binary. Being out (or possibly outed) to co-workers or one's boss, or needing assistance with documents (a tangible incongruence between one's legal identity and social identity in the hands of employers) meant supervisors and co-workers trusted with this information played key roles in determining whether or not someone was at risk for workplace mistreatment. This is consistent with findings from Lombardi [36], who found that being out to friends and coworkers was positively associated with experiences of discrimination. Also, Dietert and Dentice [37] found some participants who came out to their bosses experienced harassment and risked losing their jobs after coming out. Serano [30] points out that once someone discloses being trans (or is outed by someone else as a trans person), passing privilege is lost. Being out increases trans visibility, and out individuals can serve as role models of success for others looking to transition as well as providing a living education tool for co-workers and supervisors. But it also risks individuals losing the ability to be seen in their true gender identity.

In this study, there was a distinct gendered aspect to the experiences of workplace discrimination. Trans men were more likely to experience being misgendered and a breach of privacy, while trans women were more likely to report a wider variety of types of discrimination. Previously, Dietert and Dentice [37] found misgendering coming from both co-workers and upper management. Schilt [38] found that, particularly among cis gender heterosexual women, reactions to open transitions at work focused around bodily changes that served to delegitimize the trans men's identities as men by focusing on the body as female through the vehicles of genitalia and surgeries framed within a cis gender narrative (breast removal and hysterectomies).

Trans men may be scrutinized in the workplace for their choices regarding hormones and surgery, and view discussions about their genitalia, hormones and surgical choices as constituting a major breach of privacy regarding their personal medical information [39,40]. Furthermore, the slipping into female pronouns by co-workers is undermining and stigmatizing and suggests that those who misgender view their trans masculine co-workers as "still female"—a tangible form of differential treatment from cis gender men in the workplace [41,42]. The trans men in this study were also significantly younger than their trans female counterparts. This may have played a factor in their being so frequently misgendered. Lombardi [36] found that those who transitioned at age 30 or younger experienced greater levels of discrimination and concurrent stress. Trans men may start out being perceived as masculine women [42]. But the effects of hormones during transition for trans men often include initially having a younger, more teenager-like appearance. Masculinizing hormones lead to developing acne, going through voice change, and peach fuzz beard growth typical of teenage boys [43]. This may undermine perceptions of authority and competence. Significantly, in this study as age category increased, misgendering decreased.

The reporting by trans women of a broader range of discriminatory treatment categories as compared to trans men is consistent with the literature citing trans women facing harassment, demotion, termination, pencil-whipping (being written up for minor infractions) and heightened scrutiny for their workplace attire. A survey of legal cases brought against employers by trans women revealed that these types of mistreatment began when someone first came out and was looking to transition, and then again when employers felt threatened when these workers started wearing dresses to work and showing a more distinctly feminine physical appearance [41]. It may be explained through the lens of transmisogyny in that these employees simultaneously lose credibility as male workers while simultaneously being held up to impossible feminine standards [29,30]. In this study, age was a protective factor for trans women. The mean age of trans women in this study (42.6) put them squarely in middle-age. It is possible that they transitioned later in life and resemble the participants discussed by Schilt [44], whose later life transitions corresponded with deep investment in their careers and who either worked in a family business or stayed on post-retirement as consultants who were specialists in their fields. For these women, going stealth could never be an option [45].

Race played a distinctly intersectional role consistent with the literature, in that white participants were more likely to be shielded from privacy breach and workplace violence. This may be a reflection of white privilege, wherein being white is associated in general with less mistreatment [46]. Or it may be an embodied form of white racism such as when black bodies are automatically viewed as suspect [47]. Lombardi [36] found white transgender participants reported experiencing the lowest number of transphobic events during the year of her study. In a study by Grant et al. [5], trans people of color reported higher rates of unemployment, job loss, and abuse by police.

## 5. Limitations

The original researchers used convenience and snowball sampling for this study. By recruiting participants at a transgender leadership summit, LGBT community centers, and grassroots organizations, the study is open to self-selection bias. Since the study was sponsored by an advocacy organization, it is also open to the response biases of extreme responding (regression fallacy) and social desirability. Since the state of California had a transgender-inclusive law on the books at the

time the survey was administered, this may impact generalizability. Two questions in the survey were subjective measures. The final limitation is the relatively small sample size. Categories on some variables had to be combined, thereby losing the ability to test for subgroup differences.

Future research must take into account the changing political landscape and impingement of human rights as these relate to trans-identified employees and job seekers. Particularly in the United States, researchers should consider a more in-depth look at the role of IDs, bathroom policies, age of transition, promotion of religious freedom laws, the blocking of nondiscrimination laws, the increased cultural discussions on gender fluidity and non-binary identities, changing attitudes towards microaggressions in the workplace, as well as the role of racism in intersection with racist immigration policies under the Trump administration. In the UK and Australia, the impact of the myth of trans women as sexually predatory men in dresses (recently popularized by cis gender radical feminists) should also be examined.

## 6. Conclusions

In this study, sexual citizenship theory was used as a framework to examine employment discrimination among people who are transgender. The questions driving this study centered around how the transgender-specific factors of gender identity, passing/blending, conformity to the gender binary, being out at work and needing assistance with IDs impacted the likelihood of experiencing workplace discrimination. Of equal importance was investigating who was most likely to experience specific types of discrimination and unemployment. Theory helped frame the particular types of social exclusion or marginalization experienced by trans people to include the classic citizenship theory categories of civil, political and social citizenship, along with the impact of cisgender entitlement and transmisogyny as sexual citizenship theoretical expansions. The findings of this study address these questions, bearing out much of what has already been discussed in other studies about discrimination, while adding to the knowledge base. Participants experienced discrimination in all three realms (civil, political and social), in ways unique to trans individuals at the hands of cis gender entitlement and transmisogyny. These findings amplify the voices of trans people in bearing witness to their mistreatment and focusing on an area of which there is a dearth of evidence-based peer-reviewed research. They support the ongoing need for rights-based protections, workplace educational workshops, and redistributive policies that mitigate the impacts of poverty.

**Funding:** This research received no external funding.

**Acknowledgments:** The author wishes to thank the Transgender Law Center for providing the data for this secondary data analysis.

**Conflicts of Interest:** The author declares no conflict of interest. The Transgender Law Center had no role in the design of this secondary data analysis or interpretation of data; in the writing of the manuscript, or in the decision to publish the results.

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
