# Peer review of "Sexual Citizenship Theory and Employment Discrimination among Transgender-Identified People"

_societies, doi:10.3390/soc10010017_

Round 1

Reviewer 1 Report

Excellent paper.  The topic is of great interest, and similar papers -- rigorous empirical studies well related to a conceptual framework -- are quite rare.  The theoretical and empirical parts are both very solid.  Congratulations.  

My only suggestion is that that in Section 1.2 (Scope of the Problem) you fail to cite one of the few other rigorous empirical studies of employment discrimination against transgender people. It was published in 2010 by Make the Road New York as Transgender Need Not Apply, A Report on Gender Identity Job Discrimination.  You need to cite this paper, (and perhaps some of the other studies cited by it).  The report is available online in the Digital Commons site of the Cornell School of Industrial and Labor Relations (https://digitalcommons.ilr.cornell,edu/institutes.

Author Response

Greetings and thank you for pointing me to this study. I had heard of it through Queers for Economic Justice, but never saw the actual report.  I have added this citation, with sentences in two places:

Lines 60 - 62:

In a 2010 study by Make the Road New York [6], 52% of the trans people they surveyed reported experiencing discrimination on the job, while 49% reported receiving no employment offers while living as a trans person.

Lines 485 - 487

The 2010 matched pair testing study [6] highlighted difficulties trans people face in seeking gainful employment when qualifications are not at issue.

See updated draft for your reference and review. The changes are highlighted in red (using track changes was not an option because I could not remove my name from the sidebar notations).

Reviewer 2 Report

Ths paper presents your findings clearly, and the contents will be of some interest to non-specialist readers.

The data analysis is thorough, possibly excessively sio since the sample may not closely resemble the population, and the amount iof detail reported in the text risks 'burying' your 'headline' findings.  For myself (not a trans-specialist) these are that the respondents are well-education (86% have at least some college) buit 41% have annual incomes beneath $20K. Is this contrast a good indicator odf the cost of being non-cis? Two-thirds report some form of díscrimination at work, and this experience becomes more common when individuals are 'out' to their bosses, and among FtM than MtF. These findings may be confirming previous research, and will not surprise or excite specialists, but in Societies you will have mainly nopn-specialist readers.

Author Response

Greetings and thank you for your response. I have highlighted the relatively low income problems reported by trans people throughout the study. But in response to your comments, I added the following to the sentence beginning on line 466:

Poverty and unemployment are well-documented problems faced by members of the trans community and with 41% of respondents in this study reporting annual incomes under $20k, this sample was no exception [5, 32, 33].

It has been highlighted in the text (rather than track changes) due to my name appearing in the track changes comments. Please see attached.

Thank you for your consideration.
